# Manifold structure in graph embeddings

**Patrick Rubin-Delanchy**
University of Bristol
patrick.rubin-delanchy@bristol.ac.uk

## Abstract

Statistical analysis of a graph often starts with embedding, the process of representing its nodes as points in space. How to choose the embedding dimension is a nuanced decision in practice, but in theory a notion of true dimension is often available. In spectral embedding, this dimension may be very high. However, this paper shows that existing random graph models, including graphon and other latent position models, predict the data should live near a much lower-dimensional set. One may therefore circumvent the curse of dimensionality by employing methods which exploit hidden manifold structure.

## 1 Introduction

The hypothesis that high-dimensional data tend to live near a manifold of low dimension is an important theme of modern statistics and machine learning, often held to explain why high-dimensional learning is realistically possible [66, 6, 17, 15, 21, 47]. The object of this paper is to show that, for a theoretically tractable but rich class of random graph models, such a phenomenon occurs in the spectral embedding of a graph.

Manifold structure is shown to arise when the graph follows a latent position model [31], wherein connections are posited to occur as a function of the nodes' underlying positions in space. Because of their intuitive appeal, such models have been employed in a great diversity of disciplines, including social science [38, 45, 24], neuroscience [19, 55], statistical mechanics [37], information technology [74], biology [56] and ecology [22]. In many more endeavours latent position models are used — at least according to Definition 1 (to follow) — but are known by a different name; examples include the standard [32], mixed [2] and degree-corrected [35] stochastic block models, random geometric graphs [53], and the graphon model [42], which encompasses them.

Spectral embedding obtains a vector representation of each node by eigencomposition of the adjacency or normalised Laplacian matrix of the graph, and it is not obvious that a meaningful connection to the latent positions of a given model should exist. One contribution of this article is to make the link clear, although existing studies already take us most of the way: Tang and co-authors [65] and Lei [40] construct identical maps respectively assuming a positive-definite kernel (here generalised to indefinite) and a graphon model (here extended to $d$ dimensions). Through this connection, the notion of a true embedding dimension $D$ emerges, which is the large-sample rank of the expected adjacency matrix, and it is potentially much greater than the latent space dimension $d$.

The main contribution of this article is to demonstrate that, though high-dimensional, the data live 'close' to a low-dimensional structure — a distortion of latent space — of dimension governed by the curvature of the latent position model kernel along its diagonal. One should have in mind, as the typical situation, a $d$-dimensional manifold embedded in infinite-dimensional ambient space. However, it would be a significant misunderstanding to believe that graphon models, acting on the unit interval, so that $d = 1$, could produce only one-dimensional manifolds. Instead, common Hölder $\alpha$-smoothness assumptions on the graphon [71, 26, 40] limit the maximum possible manifold dimension to $2/\alpha$.

By 'close', a strong form of consistency is meant [14], in which the largest positional error vanishes as the graph grows, so that subsequent statistical analysis, such as manifold estimation, benefits doubly from data of higher quality, including proximity to manifold, and quantity. This is simply established by recognising that a generalised random dot product graph [58] (or its infinite-dimensional extension [40]) is operating in ambient space, and calling on the corresponding estimation theory.

It is often argued that a relevant asymptotic regime for studying graphs is *sparse*, in the sense that, on average, a node's degree should grow less than linearly in the number of nodes [48]. The afore-mentioned estimation theory holds in such regimes, provided the degrees grow quickly enough — faster than logarithmically — this rate corresponding to the information theoretic limit for strong consistency [1]. The manner in which sparsity is induced, via global scaling, though standard and required for the theory, is not the most realistic, failing the test of projectivity among other desiderata [16]. Several other papers have treated this matter in depth [50, 70, 9].

The practical estimation of $D$, which amounts to rank selection, is a nuanced issue with much existing discussion — see [55] for a pragmatic take. A theoretical treatment must distinguish the cases $D < \infty$ and $D = \infty$. In the former, simply finding a consistent estimate of $D$ has limited practical utility: appropriately scaled eigenvalues of the adjacency matrix converge to their population value, and all kinds of unreasonable rank selection procedures are therefore consistent. However, to quote [55], "any quest for a universally optimal methodology for choosing the "best" dimension [...], in general, for finite $n$, is a losing proposition". In the $D = \infty$ case, reference [40] finds appropriate rates under which to let $\hat{D} \to \infty$, to achieve consistency in a type of Wasserstein metric. Unlike the $D < \infty$ case, stronger consistency, i.e., in the largest positional error, is not yet available. All told, the method by [75], which uses a profile-likelihood-based analysis of the scree plot, provides a practical choice and is easily used within the R package 'igraph'. The present paper's only addition to this discussion is to observe that, under a latent position model, rank selection targets *ambient* rather than *intrinsic* dimension, whereas the latter may be more relevant for estimation and inference. For example, one might legitimately expect certain graphs to follow a latent position model on $\mathbb{R}^3$ (connectivity driven by physical location) or wish to test this hypothesis. Under assumptions set out in this paper (principally Assumption 2 with $\alpha = 1$), the corresponding graph embedding should concentrate about a three-dimensional manifold, whereas the ambient dimension is less evident, since it corresponds to the (unspecified) kernel's rank.

The presence of manifold structure in spectral embeddings has been proposed in several earlier papers, including [54, 5, 68], and the notions of model complexity versus dimension have also previously been disentangled [55, 73, 52]. That low-dimensional manifold structure arises, more generally, under a latent position model, is to the best of our knowledge first demonstrated here.

The remainder of this article is structured as follows. Section 2 defines spectral embedding and the latent position model, and illustrates this paper's main thesis on their connection through simulated examples. In Section 3, a map sending each latent position to a high-dimensional vector is defined, leading to the main theorem on the preservation of intrinsic dimension. The sense in which spectral embedding provides estimates of these high-dimensional vectors is discussed Section 4. Finally, Section 5 gives examples of applications including regression, manifold estimation and visualisation. All proofs, as well as auxiliary results, discussion and figures, are relegated to the Supplementary Material.

## 2 Definitions and examples

**Definition 1** (Latent position model). Let $f : \mathcal{Z} \times \mathcal{Z} \to [0, 1]$ be a symmetric function, called a kernel, where $\mathcal{Z} \subseteq \mathbb{R}^d$. An undirected graph on $n$ nodes is said to follow a latent position network model if its adjacency matrix satisfies

$$\mathbf{A}_{ij} \mid Z_1, \ldots, Z_n \overset{ind}{\sim} \text{Bernoulli} \{f(Z_i, Z_j)\}, \quad \text{for } i < j,$$

where $Z_1, \ldots, Z_n$ are independent and identically distributed replicates of a random vector $Z$ with distribution $F_Z$ supported on $\mathcal{Z}$. If $\mathcal{Z} = [0, 1]$ and $F_Z = \text{uniform}[0, 1]$, the kernel $f$ is known as a graphon.

Outside of the graphon case, where $f$ is usually estimated nonparametrically, several parametric models for $f$ have been explored, including $f(x, y) = \text{logistic}(\alpha - \|x - y\|)$ [31], where $\|\cdot\|$ is any

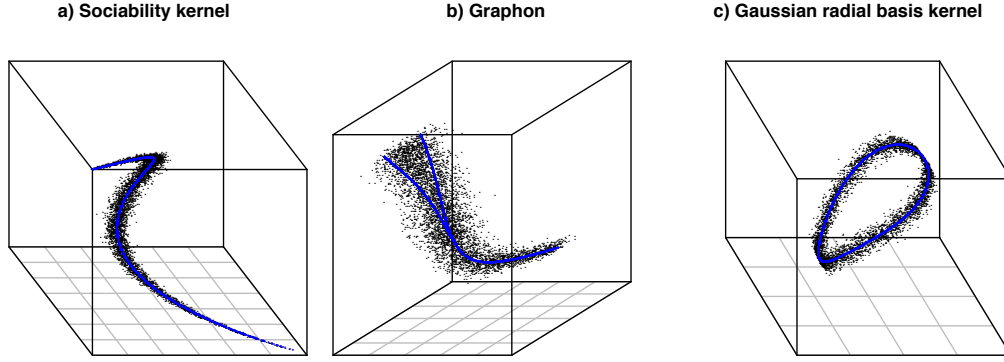

**a) Sociability kernel**  **b) Graphon**  **c) Gaussian radial basis kernel**

Figure 1: Spectral embedding of graphs simulated from three different latent position network models of latent dimension one ($d = 1$). The true ambient dimension is infinite ($D = \infty$) and only the first three coordinates are shown. The theory predicts the data should live close to a one-dimensional structure, informally called a manifold (and obviously not strictly so in example b), shown in blue. Further details in main text.

norm and $\alpha$ a parameter, $f(x, y) = \text{logistic}(\alpha + x_1 + y_1 + x_2^\top y_2)$ where $x = (x_1, x_2), y = (y_1, y_2)$ [30], $f(x, y) = \Phi(\alpha + x^\top \Lambda y)$, where $\Lambda$ is diagonal (but not necessarily non-negative) [29], the Gaussian radial basis kernel $f(x, y) = \exp\{-\|x - y\|^2/(2\sigma^2)\}$, and finally the sociability kernel $f(x, y) = 1 - \exp(-2xy)$ [3, 49, 8, 69, 16] where, in the last case, $x, y \in \mathbb{R}_+$. Typically, such functions have infinite rank, as defined in Section 3, so that the true ambient dimension is $D = \infty$.

**Definition 2** (Adjacency and Laplacian spectral embedding)**.** Given an undirected graph with adjacency matrix $\mathbf{A}$ and a finite embedding dimension $\hat{D} > 0$, consider the spectral decomposition $\mathbf{A} = \hat{\mathbf{U}}\hat{\mathbf{S}}\hat{\mathbf{U}}^\top + \hat{\mathbf{U}}_\perp \hat{\mathbf{S}}_\perp \hat{\mathbf{U}}_\perp^\top$, where $\hat{\mathbf{S}}$ is the $\hat{D} \times \hat{D}$ diagonal matrix containing the $\hat{D}$ largest-by-magnitude eigenvalues of $\mathbf{A}$ on its diagonal, and the columns of $\hat{\mathbf{U}}$ are corresponding orthonormal eigenvectors. Define the adjacency spectral embedding of the graph by $\hat{\mathbf{X}} = [\hat{X}_1|\ldots|\hat{X}_n]^\top := \hat{\mathbf{U}}|\hat{\mathbf{S}}|^{1/2} \in \mathbb{R}^{n \times \hat{D}}$. Define its Laplacian spectral embedding by $\check{\mathbf{X}} = [\check{X}_1|\ldots|\check{X}_n]^\top := \check{\mathbf{U}}|\check{\mathbf{S}}|^{1/2}$ where $\check{\mathbf{U}}$, $\check{\mathbf{S}}$ are instead obtained from the spectral decomposition of $\mathbf{L} := \mathbf{D}^{-1/2}\mathbf{A}\mathbf{D}^{-1/2}$ and $\mathbf{D}$ is the diagonal degree matrix.

Figure 1 shows point clouds obtained by adjacency spectral embedding graphs from three latent position network models on $\mathbb{R}$ (with $n = 5000$, $\hat{D} = 3$). The first was generated using the kernel $f(x, y) = 1 - \exp(-2xy)$, and latent positions $Z_i \overset{i.i.d.}{\sim} \text{Gamma}(1, 1)$, truncated to a bounded set, suggested to represent node "sociability" [16]. The second corresponds to a graphon model with kernel chosen to induce filamentary latent structure with a branching point. In the third, the Gaussian radial basis kernel $\exp\left\{-\|x - y\|_\circ^2/(2\sigma^2)\right\}$ is used, with $Z_i$ uniformly distributed on a circle and $\|x - y\|_\circ$ representing geodesic distance. In all three cases the true ambient dimension is infinite, so that the figures show only a 3-dimensional projection of the truth. Our main result, Theorem 3, predicts that the points should live close to a one-dimensional structure embedded in infinite dimension, and this structure (or rather its three-dimensional projection) is shown in blue.

## 3 Spectral representation of latent position network models

In this section, a map $\phi : \mathbb{R}^d \to L^2(\mathbb{R}^d)$ is defined that transforms the latent position $Z_i$ into a vector $X_i := \phi(Z_i)$. The latter is shown to live on a low-dimensional subset of $L^2(\mathbb{R}^d)$, informally called a manifold, which is sometimes but by no means always of dimension $d$. As suggested by the notation and demonstrated in the next section, the point $\hat{X}_i$, obtained by spectral embedding, provides a consistent estimate of $X_i$ and lives near (but not on) the manifold.

Assume that $f \in L^2(\mathbb{R}^d \times \mathbb{R}^d)$, if necessary extending $f$ to be zero outside its original domain $\mathcal{Z}$, though a strictly stronger (trace-class) assumption is to come. Define an operator $A$ on $L^2(\mathbb{R}^d)$

through

$$Ag(x) = \int_{\mathbb{R}^d} f(x,y)g(y)dy, \quad g \in L^2(\mathbb{R}^d).$$

Since $\int_{\mathbb{R}^d} \int_{\mathbb{R}^d} |f(x,y)|^2 dx\, dy < \infty$, the function $f$ is a Hilbert-Schmidt kernel and the operator $A$ is compact. The said 'true' ambient dimension $D$ is the rank of $f$, that is, the potentially infinite dimension of the range of $A$, but this plays no role until the next section.

From the symmetry of $f$, the operator is also self-adjoint and therefore there exists a possibly infinite sequence of non-zero eigenvalues $\lambda_1 \geq \lambda_2 \geq \ldots$ with corresponding orthonormal eigenfunctions $u_1, u_2, \ldots \in L^2(\mathbb{R}^d)$ such that

$$Au_j = \lambda_j u_j.$$

Moreover, $(u_j)$ and $(\lambda_j)$ can be extended, by the potential addition of further orthonormal functions, which give an orthonormal basis for the null space of $A$ and are assigned the eigenvalue zero, so that $(u_j)$ provides a complete orthonormal basis of $L^2(\mathbb{R}^d)$ and, for any $g \in L^2(\mathbb{R}^d)$,

$$g = \sum_j \langle g, u_j \rangle u_j \quad \text{and} \quad Ag = \sum_j \lambda_j \langle g, u_j \rangle u_j.$$

The notions employed above are mainstream in functional analysis, see e.g. [18], and were used to analyse graphons in [34, 8, 42, 72, 40].

Define the operator $|A|^{1/2}$ through the spectral decomposition of $A$ via [39, p.320]

$$|A|^{1/2} g = \sum_j |\lambda_j|^{1/2} \langle g, u_j \rangle u_j.$$

**Assumption 1.** $A$ is trace-class [39, p.330], that is,

$$\mathrm{tr}(A) = \sum_j |\lambda_j| < \infty.$$

This assumption, common in operator theory, was employed by [40] to draw the connection between graphon estimation and spectral embedding. Under Assumption 1, the operator $|A|^{1/2}$ is Hilbert-Schmidt, having bounded Hilbert-Schmidt norm

$$\||A^{1/2}\|_{\mathrm{HS}}^2 = \mathrm{tr}(A),$$

and representing $|A|^{1/2}$ as an integral operator

$$|A|^{1/2} g(x) = \int_{\mathbb{R}^d} k(x,y)g(y)dy,$$

the kernel $k(x,y) := \sum_j |\lambda_j|^{1/2} u_j(x) u_j(y)$ is an element of $L^2(\mathbb{R}^d \times \mathbb{R}^d)$, using the identity

$$\|A^{1/2}\|_{\mathrm{HS}}^2 = \int_{\mathbb{R}^d} \int_{\mathbb{R}^d} |k(x,y)|^2 dy\, dx.$$

By explicitly partitioning the image of the outer integrand (as is implicitly done in a Lebesgue integral), the Lebesgue measure of intervals of $x$ where $\int_{\mathbb{R}^d} |k(x,y)|^2 dy = \infty$ is determined to be zero, and therefore $k(x,\cdot) \in L^2(\mathbb{R}^d)$ almost everywhere in $\mathbb{R}^d$.

Note that $k(x,\cdot)$ has coordinates $(|\lambda_j|^{1/2} u_j(x))$ with respect to the basis $(u_j)$. Moreover, under the indefinite inner product

$$[g,h] := \sum_j \mathrm{sgn}(\lambda_j) \langle g, u_j \rangle \langle h, u_j \rangle, \quad g, h \in L^2(\mathbb{R}^d),$$

we have

$$\int_{\mathbb{R}^d} [k(x,\cdot), k(y,\cdot)] g(y) dy = \int_{\mathbb{R}^d} \sum_j \lambda_j u_j(x) u_j(y) g(y) dy$$
$$= \sum_j \lambda_j \langle g, u_j \rangle u_j(x)$$
$$= Ag(x),$$

and therefore $[k(x, \cdot), k(y, \cdot)] = f(x, y)$ almost everywhere, that is, with respect to Lebesgue measure on $\mathbb{R}^d \times \mathbb{R}^d$.

The map $\phi : x \mapsto k(x, \cdot)$, whose precise domain is discussed in the proof of Theorem 3, transforms each latent position $Z_i$ to a vector $X_i := \phi(Z_i)$. (That $\hat{X}_i$ provides an estimate of $X_i$ will be discussed in the next section.) It is now demonstrated that $X_i$ lives on a low-dimensional manifold in $L^2(\mathbb{R}^d)$.

The operator $A$ admits a unique decomposition into positive and negative parts satisfying [18, p. 213]

$$A = A_+ - A_-, \; A_+ A_- = 0, \; \langle A_+ g, g \rangle \geq 0, \; \langle A_- g, g \rangle \geq 0, \quad \text{for all } g \in L^2(\mathbb{R}^d).$$

These parts are, more explicitly, the integral operators associated with the symmetric kernels

$$f_+(x, y) := \sum_j \max(\lambda_j, 0) u_j(x) u_j(y), \quad f_-(x, y) := \sum_j \max(-\lambda_j, 0) u_j(x) u_j(y),$$

so that $f = f_+ - f_-$ in $L^2(\mathbb{R}^d \times \mathbb{R}^d)$.

**Assumption 2.** There exist constants $c > 0$ and $\alpha \in (0, 1]$ such that for all $x, y \in \mathcal{Z}$,

$$\Delta^2 f_+(x, y) \leq c \|x - y\|_2^{2\alpha}; \quad \Delta^2 f_-(x, y) \leq c \|x - y\|_2^{2\alpha},$$

where $\Delta^2 f.(x, y) := f.(x, x) + f.(y, y) - 2f.(x, y)$.

To give some intuition, this assumption controls the curvature of the kernel, or its operator-sense absolute value, along its diagonal. In the simplest case where $f$ is positive-definite and $\mathcal{Z} = \mathbb{R}$, the assumption is satisfied with $\alpha = 1$ if $\partial^2 f/(\partial x \partial y)$ is bounded on $\{(x, y) : x, y \in \mathcal{Z}, |x - y| \leq \epsilon\}$, for some $\epsilon > 0$. The assumption should not be confused with a Hölder continuity assumption, as is common with graphons, which would seek to bound $|f(x, y) - f(x', y')|$ [26]. Nevertheless, such an assumption can be exploited to obtain some $\alpha \leq 1/2$, as discussed in Example 2.

**Assumption 3.** The distribution of $Z$ is absolutely continuous with respect to $d$-dimensional Lebesgue measure.

Since it has always been implicit that $d \geq 1$, the assumption above does exclude the stochastic block model, typically represented as a latent position model with discrete support. This case is inconvenient, although not insurmountably so, to incorporate within a proof technique which works with functions defined *almost* everywhere, since one must verify that the vectors representing communities do not land on zero-measure sets where the paper's stated identities might not hold. However, the stochastic block model is no real exception to the main message of this paper: in this case the relevant manifold has dimension zero.

The vectors $X_1 = k(Z_1, \cdot), \ldots, X_n = k(Z_n, \cdot)$ will be shown to live on a set of low *Hausdorff* dimension. The $a$-dimensional Hausdorff content [7] of a set $\mathcal{M}$ is

$$\mathscr{H}^a(\mathcal{M}) = \inf \left\{ \sum_i |S_i|^a : \mathcal{M} \subseteq \bigcup_i S_i \right\},$$

where $S_i$ are arbitrary sets, $|S|$ denotes the diameter of $S$, here to be either in the Euclidean or $L^2$ norm, as the situation demands. The Hausdorff dimension of $\mathcal{M}$ is

$$\dim(\mathcal{M}) = \inf \left\{ a : \mathscr{H}^a(\mathcal{M}) = 0 \right\}.$$

**Theorem 3.** *Under Assumptions 1–3, there is a set $\mathcal{M} \subset L^2(\mathbb{R}^d)$ of Hausdorff dimension $d/\alpha$ which contains $k(Z, \cdot)$ with probability one.*

**Example 1** (Gaussian radial basis function)**.** For arbitrary finite $d$, consider the kernel

$$f(x, y) = \exp \left( -\frac{\|x - y\|_2^2}{2\sigma^2} \right), \quad x, y \in \mathcal{Z} \subseteq \mathbb{R}^d,$$

known as the Gaussian radial basis function. Since $f$ is positive-definite, the trace formula [11]

$$\text{tr}(A) = \int_x f(x, x) dx$$

shows it is trace-class if and only if $\mathcal{Z}$ is bounded. Assume $F_Z$ is absolutely continuous on $\mathcal{Z}$. Since $1 - \exp(a^2) \le a^2$ for $a \in \mathbb{R}$, it follows that

$$f(x,x) - f(x,y) \le \frac{1}{2\sigma^2}\|x - y\|_2^2,$$

and since $f_+ = f, f_- = 0$ (because $f$ is positive-definite)

$$\Delta^2 f_+ \le \frac{1}{\sigma^2}\|x - y\|_2^2; \quad \Delta^2 f_- = 0,$$

satisfying Assumption 2 with $\alpha = 1$. The implication of Theorem 3 is that *the vectors $X_1, \ldots, X_n$ (respectively, their spectral estimates $\hat{X}_1, \ldots, \hat{X}_n$) live on (respectively, near) a set of Hausdorff dimension $d$.* △

**Example 2** (Smooth graphon)**.** If a graphon $f$ is Hölder continuous for some $\beta \in (1/2, 1]$, that is,

$$|f(x,y) - f(x,y')| \le c|y - y'|^\beta$$

for some $c > 0$, then it is trace-class [40], [39, p.347].

If, additionally, it is positive-definite, it can be shown that Assumption 2 holds with $\alpha = 1$ if the partial derivative $\partial^2 f/(\partial x \partial y)$ is bounded in a neighbourhood of $\{(x,x) : x \in [0,1]\}$. Then the implication of Theorem 3 is that *the vectors $X_1, \ldots, X_n$ (respectively, their spectral estimates $\hat{X}_1, \ldots, \hat{X}_n$) live on (respectively, near) a set of Hausdorff dimension one*. In visual terms, they may cluster about a curve or filamentary structure (see Figure 1b).

On the other hand, if $f$ is only Hölder continuous ($\beta \in (1/2, 1]$) and positive-definite, Assumptions 1–3 hold still, with $\alpha = \beta/2$. In this case, the supporting manifold has dimension not exceeding $2/\beta$, and is at most two-dimensional if $f$ is Lipschitz ($\beta = 1$). △

# 4 Consistency of spectral embedding

The sense in which the spectral embeddings $\hat{X}_i, \breve{X}_i$ provide estimates of $X_i = k(Z_i, \cdot)$ is now discussed. The strongest guarantees on concentration about the manifold can be given if $f \in L^2(\mathbb{R}^d \times \mathbb{R}^d)$ has finite rank, that is, the sequence of non-zero eigenvalues of the associated integral operator $A$ has length $D < \infty$. Existing infinite rank results are discussed in the Supplementary Material. Random graph models where the finite rank assumption holds include:

1. all standard [32], mixed [2] and degree-corrected [35] stochastic block models — proofs of which can be found in [57, 63, 41, 59, 58, 40];

2. all latent position models with polynomial kernel of finite degree (Lemma 4, Supplementary Material).

3. under sparsity assumptions, latent position models with analytic kernel (Lemma 5, Supplementary Material).

The second of these items seems particularly useful, perhaps taken with the possibility of Taylor approximation.

Under a finite rank assumption, the high-dimensional vectors $X_i$ will be identified by their coordinates with respect to the basis $(u_j)$, truncated to the first $D$, since the remaining are zero. The latent position model of Definition 1 then becomes:

$$\mathbf{A}_{ij} \mid X_1, \ldots, X_n \overset{ind}{\sim} \text{Bernoulli}\left\{X_i^\top \mathbf{I}_{p,q} X_j\right\}, \quad \text{for } i < j,$$

where $\mathbf{I}_{p,q}$ is the diagonal matrix consisting of $p$ +1's (for as many positive eigenvalues) followed by $q$ −1's (for as many negative) and $p + q = D$. The model of a generalised random dot product graph [58] is recognised, from which strong consistency is established [58, Theorems 5,6]:

$$\max_{i \in [n]} \|\mathbf{Q}\hat{X}_i - X_i\|_2 = O_\mathbb{P}\left(\frac{(\log n)^c}{n^{1/2}}\right), \quad \max_{i \in [n]} \left\|\tilde{\mathbf{Q}}\breve{X}_i - \frac{X_i}{\left[X_i, \sum_j X_j\right]^{1/2}}\right\|_2 = O_\mathbb{P}\left(\frac{(\log n)^c}{n\rho_n^{1/2}}\right),$$

for some $c > 0$, where $\mathbf{Q}, \tilde{\mathbf{Q}}$ are random unidentifiable matrices belonging to the indefinite orthogonal group $\mathbb{O}(p, q) = \{\mathbf{M} \in \mathbb{R}^{D \times D} : \mathbf{M}\mathbf{I}_{p,q}\mathbf{M}^\top = \mathbf{I}_{p,q}\}$, and $\rho_n$ is the graph sparsity factor, assumed to satisfy $n\rho_n = \omega\{(\log n)^{4c}\}$. Moreover, $\mathbf{Q}^{-1}$ has almost surely bounded spectral norm [62], from which the consistency of many subsequent statistical analyses can be demonstrated. To give a generic argument, assume the method under consideration returns a set, such as a manifold estimate, and can therefore be viewed as a function $S : \mathbb{R}^{n \times D} \to \mathcal{P}(\mathbb{R}^D)$ (where $\mathcal{P}(A)$ is the power set of a set $A$). If $S$ is Lipschitz continuous in the following sense:

$$d_H\{S(\mathbf{X}), S(\mathbf{Y})\} \leq c \max_{i \in [n]} \|X_i - Y_i\|_2,$$

where $c > 0$ is independent of $n$ and $d_H$ denotes Hausdorff distance, then

$$d_H\{S(\hat{\mathbf{X}}), S(\mathbf{Q}^{-1}\mathbf{X})\} \leq c \, \|\mathbf{Q}^{-1}\| \max_{i \in [n]} \|\mathbf{Q}\hat{X}_i - X_i\|_2 \to 0,$$

with high probability. One must live with never knowing $\mathbf{Q}$ and fortunately many inferential questions are unaffected by a joint linear transformation of the data, some examples given in [58].

A refinement of this argument was used to prove consistency of the kernel density estimate and its persistence diagram in [62], and more have utilised the simpler version with $\mathbf{Q} \in \mathbb{O}(D)$, under positive-definite assumptions [43, 59, 5, 68].

# 5 Applications

## 5.1 Graph regression

Given measurements $Y_1, ..., Y_m \in \mathbb{R}$ for a subset $\{1, \ldots, m\}$ of the nodes, a common problem is to predict $Y_i$ for $i > m$. After spectral embedding, each $\hat{X}_i$ can be treated as a vector of covariates — a feature vector — but by the afore-going discussion their dimension $D$ (or any sensible estimate thereof) may be large, giving a potentially false impression of complexity. On the basis of recent results on the generalisation error of neural network regression under low intrinsic (Minkowski) dimension [47], one could hope for example that spectral embedding, with $\hat{D} \to \infty$ appropriately slowly, followed by neural network regression, could approach the rate $n^{-2\beta/(2\beta+d)}$ rather than the standard non-parametric rate $n^{-2\beta/(2\beta+D)}$ (with $\beta$ measuring regression function smoothness).

For illustration purposes, consider a linear regression model $Y_i = a + bZ_i + \epsilon_i$ where, rather than $Z_i$ directly, a graph posited to have latent positions $Z_i \in \mathbb{R}$ is observed, with unknown model kernel. Under assumptions set out in this paper (with $\alpha = 1$), the spectral embedding should concentrate about a one-dimensional structure, and the performance of a nonlinear regression technique that exploits hidden manifold structure may come close to that of the ideal prediction based on $Z_i$.

For experimental parameters $a = 5$, $b = 2$, $\epsilon \sim \text{normal}(0, 1)$, the kernel $f(x, y) = 1 - \exp(-2xy)$ (seen earlier), $\hat{D} = 100$ and $n = 5000$, split into 3,000 training and 2,000 test examples, the out-of-sample mean square error (MSE) of four methods are compared: a feedforward neural network (using default R keras configuration with obvious adjustments for input dimension and loss function; MSE 1.25); the random forest [10] (default configuration of the R package randomForest; MSE 1.11); the Lasso [67] (default R glmnet configuration; MSE 1.58); and least-squares (MSE 1.63). In support of the claim, the random forest and the neural network reach out-of-sample MSE closest to the ideal rate of 1. A graphical comparison is shown in Figure 3 (Supplementary Material).

## 5.2 Manifold estimation

Obtaining a manifold estimate has several potential uses, including gaining insight into the dimension and kernel of the latent position model. A brief experiment exploring the performance of a provably consistent manifold estimation technique is now described. Figure 4 (Supplementary Material) shows the one-dimensional kernel density ridge [51, 27] of the three point clouds studied earlier (Section 2, Figure 1), obtained using the R package ks in default configuration. The quality of this (and any) manifold estimate depends on the degree of off-manifold noise, for which asymptotic control has been discussed, and manifold structure, where much less is known under spectral embedding. The estimate is appreciably worse in the second case although, in its favour, it correctly detects a branching point which is hard to distinguish by eye.

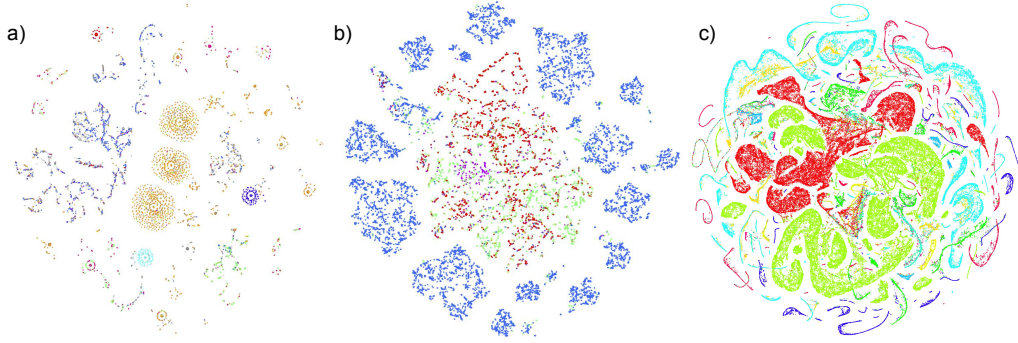

Figure 2: Non-linear dimension reduction of spectral embeddings. a) Graph of computer-to-computer network flow events on the Los Alamos National Laboratory network, from the publically available dataset [36], with colours indicating port preference (intrinsic dimension estimates 2.5/1.8/3.9); b) graph of computer-to-computer authentication events on the same network, with colours indicating authentication type (intrinsic dimension estimates 1.6/1.3/2.4); c) graph of consumer-restaurant ratings, showing only the restaurants, extracted from the publically available Yelp dataset, with colours indicating US state (intrinsic dimension estimates 4.1/3.9/3.0). In each case, the graph is embedded into $\mathbb{R}^{10}$, followed by applying t-distributed stochastic neighbour embedding [44].

### 5.3   Real data

From an exploratory data analysis perspective, the theory presented helps make sense of spectral embedding followed by non-linear dimension reduction, an approach that seems to generate informative visualisations of moderate-size graphs at low programming and computational cost. This is illustrated in three real data examples, detailed in Figure 2. In each case, the graph is embedded into $\mathbb{R}^{10}$, followed by applying t-distributed stochastic neighbour embedding [44]. Other methods such as Uniform manifold approximation [46] were tested with comparable results (Supplementary Material, Figure 5), but directly spectrally embedding into $\mathbb{R}^2$, i.e., linear dimension reduction, is less effective (Supplementary Material, Figure 6). Because the first example is reproduced from [58], that paper's original choice, $\hat{D} = 10$, although avowedly arbitrary, is upheld. The method of [75], advocated in the introduction, instead returns a rank estimate $\hat{D} = 6$.

In the first example, the full graph of connections between computers on the Los Alamos National Laboratory network, with roughly 12 thousand nodes and one hundred thousand edges, is curated from the publically available release [36]. The colours in the plot indicate the node's most commonly used port (e.g. 80=web, 25=email), reflecting its typical communication behaviour (experiment reproduced from [58]). The second example concerns the full graph of authentications between computers on the same network (a different dataset from the same release), with roughly 18,000 nodes and 400,000 edges, colours now indicating authentication type. In the third example a graph of user-restaurant ratings is curated from the publically available Yelp dataset, with over a million users, 170,000 restaurants and 5 million edges, where an edge $(i, j)$ exists if user $i$ rated restaurant $j$. The plot shows the embedding of restaurants, coloured by US state. It should be noted that in every case the colours are independent of the embedding process, and thus loosely validate geometric features observed. Estimates of the instrinsic dimensions of each of the point clouds are also included. Following [47] we report estimates obtained using local principal component analysis [25, 12], maximum likelihood [28] and expected simplex skewness [33], in that order, as implemented in the R package 'intrinsicDimension'.

## 6   Conclusion

This paper gives a model-based justification for the presence of hidden manifold structure in spectral embeddings, supporting growing empirical evidence [54, 5, 68]. The established correspondence between the model and manifold dimensions suggests several possibilities for statistical inference. For example, determining whether a graph follows a latent position model on $\mathbb{R}^d$ (e.g. connectivity driven by physical location, when $d = 3$) could plausibly be achieved by testing a manifold hypothesis

[21], that targets the intrinsic dimension, $d$, and *not* the more common low-rank hypothesis [61], that targets the ambient dimension, $D$. On the other hand, the description of the manifold given in this paper is by no means complete, since the only property established about it is its Hausdorff dimension. This leaves open what other inferential possibilities could arise from investigating other manifold properties, such as smoothness or topological structure.

## Broader Impact

*Beneficiaries.* This work gives a new perspective on graph embedding, which will first benefit network science research, in providing a new paradigm in which to explore, compare and test different latent position network models. The work will also benefit applied data science in demystifying the presence of 'manifold-like' features appearing in embeddings, the practice of non-linear dimension reduction after spectral embedding (examples in Section 5.3), and in general suggesting the incorporation of manifold learning in supervised and unsupervised inference problems based on graphs. Finally, researchers in manifold learning may enjoy this new application domain.

*Who may be put at a disadvantage.* The growth of network science is fuelled by the internet [48] and the chief ethical concern with research in this area is privacy. All data used here are in the public domain, and those from Los Alamos National Laboratory are anonymised and approved by the organisations' Human Subject review board.

*Consequence of failure of system.* Not applicable.

*Leveraging biases in the data.* Of a plethora of graph embedding techniques, spectral embedding has arguably benefitted from the most rigorous statistical examination [57, 41, 63, 60, 64, 4, 43, 13, 14]. Nevertheless, mathematical bias is known to persist, and there is no question that bias in the usual sense will too. This should be noted, for example, when implementing spectral embedding for prediction. There are in general worrying implications in making decisions affecting humans based on network data but, as the current pandemic shows, such decisions are no less necessary and should be informed by best practice.

## Acknowledgments and Disclosure of Funding

The author would like to thank Carey Priebe for inspiring conversations and insights, and acknowledge the Heilbronn Institute for Mathematical Research and the Alan Turing Institute for direct financial support of this work.

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
