[Supplementary Material]

# Supplementary material for "Manifold structure in graph embeddings"

*Proof of Theorem 3.* We have

$$\int_{\mathbb{R}^d} \langle k(x,\cdot), k(y,\cdot) \rangle g(y) dy = \int_{\mathbb{R}^d} \sum_j |\lambda_j| u_j(x) u_j(y) g(y) dy = |A| g(y)$$

$$= \int_{\mathbb{R}^d} [f_+(x,y) + f_-(x,y)] g(y) dy,$$

and therefore

$$\langle k(x,\cdot), k(y,\cdot) \rangle = f_+(x,y) + f_-(x,y) \tag{1}$$

almost everywhere. Explicitly, there exists $N \subset \mathbb{R}^d \times \mathbb{R}^d$ with Lebesgue measure zero such that the two are equal in $\mathbb{R}^d \times \mathbb{R}^d \setminus N$. The set $N_0 = \{x \in \mathbb{R}^d : (x,y) \in N \text{ or } (y,x) \in N\}$ has $d$-dimensional Lebesgue measure 0 and, by absolute continuity of $Z$, the set $\underline{\mathcal{Z}} := \mathcal{Z} \setminus N_0$ has Hausdorff dimension $d$, satisfies $\mathbb{P}(Z \in \underline{\mathcal{Z}}) = 1$, and Equation (1) holds for all $x, y \in \underline{\mathcal{Z}}$.

Let $\mathcal{M}$ be the image of $\underline{\mathcal{Z}}$ under the map $\phi$, which in general maps any $x \in \underline{\mathbb{R}^d} := \mathbb{R}^d \setminus N_0$ to an element of $L^2(\mathbb{R}^d)$. Any cover $\underline{\mathcal{Z}} \subseteq \cup_i R_i$ with sets in $\underline{\mathbb{R}^d}$ gives a cover $\mathcal{M} \subseteq \cup_i S_i$ with sets in $L^2(\mathbb{R}^d)$, where each $S_i$ is the image of $R_i$ under $\phi$. Note that the Hausdorff dimension of $\underline{\mathcal{Z}}$ is not increased when considering only such covers.

We have

$$\|k(x,\cdot) - k(y,\cdot)\|_2^2 = \langle k(x,\cdot), k(x,\cdot) \rangle + \langle k(y,\cdot), k(y,\cdot) \rangle - 2\langle k(x,\cdot), k(y,\cdot) \rangle$$

$$= \Delta^2 f_+(x,y) + \Delta^2 f_-(x,y)$$

$$\leq c\|x - y\|_2^{2\alpha},$$

and therefore $|S_i| \leq c^{1/2} |R_i|^\alpha$ for each $i$. Hence, $\mathscr{H}^a(\mathcal{M}) \leq c^{a/2} \mathscr{H}^{a\alpha}(\underline{\mathcal{Z}})$ and $\dim(\mathcal{M}) \leq d/\alpha$. $\quad\square$

**Lemma 4.** *Consider a polynomial kernel over a bounded region $\mathcal{Z} \subset \mathbb{R}^d$,*

$$f(x,y) = \sum_{|\alpha|+|\beta|<k} c(\alpha,\beta) x^\alpha y^\beta, \quad x, y \in \mathcal{Z},$$

*where $c(\alpha,\beta) = c(\beta,\alpha) \in \mathbb{R}$, and multi-index notation is used [23], that is, $x^\alpha := \prod x^{\alpha_i}, |\alpha| := \sum \alpha_i$, for $\alpha \in \mathbb{R}^d, \alpha_i \geq 0$. The associated integral operator has finite rank.*

*Proof.* Since

$$Ag(x) = \sum_{|\alpha|<k} x^\alpha \sum_{|\alpha|+|\beta|<k} c(\alpha,\beta) \int_{\mathcal{Z}} y^\beta g(y) dy,$$

the function $Ag$ is a linear combination of a finite set of $L^2$ functions $\{x^\alpha : |\alpha| < k\}$ so that, having finite-dimensional range, the operator $A$ has by definition finite rank [18]. $\quad\square$

**Lemma 5.** *Suppose $f$ is analytic on $\mathcal{Z}$ with*

$$f(x,y) = \sum_{|\alpha|+|\beta|=1}^{\infty} c(\alpha,\beta) x^\alpha y^\beta, \quad x, y \in \mathcal{Z},$$

*i.e., no constant term, and consider a sparse graph regime $Z_i = r_n W_i$ where $W_i \overset{i.i.d.}{\sim} F_W$ and the positive scalar sequence $r_n \to 0$. Assume that $r_n = o(n^{-2/k})$, for some $k \in \mathbb{N}$, and that*

$$f_k(x,y) := \sum_{1 \leq |\alpha|+|\beta|<k} c(\alpha,\beta) x^\alpha y^\beta \in [0,1].$$

*A latent position model with kernel $f$ is asymptotically indistinguishable from one with (finite rank) kernel $f_k$.*

*Proof.* The graph adjacency matrix $\mathbf{A}$ can be coupled with another random matrix $\mathbf{A}^{(k)} \in \{0,1\}^{n \times n}$ so that

$$\mathbb{P}(\mathbf{A}_{ij} \neq \mathbf{A}_{ij}^{(k)} \mid Z_i, Z_j) = |f(Z_i, Z_j) - f_k(Z_i, Z_j)|,$$

and $\mathbf{A}^{(k)}$ marginally follows a latent position model with kernel $f_k$. Therefore,

$$\mathbb{P}(\mathbf{A}_{ij} \neq \mathbf{A}_{ij}^{(k)}) = \mathbb{E}|f(Z_i, Z_j) - f_k(Z_i, Z_j)|,$$

and

$$\begin{aligned}
\mathbb{P}(\mathbf{A} \neq \mathbf{A}^{(k)}) &\leq n^2 \mathbb{E}|f(Z_i, Z_j) - f_k(Z_i, Z_j)| \\
&= n^2 \mathbb{E}\left| \sum_{|\alpha|+|\beta|=k}^{\infty} c(\alpha, \beta) Z_i^\alpha Z_j^\beta \right| \\
&\leq n^2 \left| \sum_{|\alpha|+|\beta|=k}^{\infty} r_n^{|\alpha|+|\beta|} \right| \mathbb{E}\left| \sum_{|\alpha|+|\beta|=k}^{\infty} c(\alpha, \beta) W_i^\alpha W_j^\beta \right| \\
&= n^2 \frac{(2dr_n)^k}{1-2dr_n} \mathbb{E}|f(W_i, W_j) - f_k(W_i, W_j)|,
\end{aligned}$$

which, by the boundedness of $f$ and $f_k$, tends to zero when $n^2 r_n^k \to 0$. Therefore $\mathbf{A}$ is asymptotically indistinguishable from a graph with finite kernel rank. $\square$

### Limitations of existing infinite rank results

Under a graphon model ($d = 1$, $\mathcal{Z} = [0,1]$, $F_Z = \text{uniform}[0,1]$) with suitably growing $D$, [40] proves consistency of $\hat{X}_1, \ldots, \hat{X}_n$, in the orthogonal Wasserstein distance

$$d(\hat{F}_X, F_X) = \inf_\nu \inf_W \mathbb{E}\|W\hat{X} - X\|_2,$$

where $\hat{F}_X$ is the empirical distribution of $\hat{X}_1, \ldots, \hat{X}_n$, the pair $\hat{X}, X$ are jointly distributed as $\nu$ chosen among all distributions with respective marginals $\hat{F}_X$ and $F_X$ (the distribution of $X$ induced by $F_Z$ under the map $\phi$), and finally $W$ is some *orthogonal* transformation satisfying $[Wg, Wh] = [g, h]$ for all $g, h \in L^2([0,1])$ (for the indefinite inner product given in Section 3).

While an extension to the $d$-dimensional case is conceivable, this line of analysis is complicated in the present context by two issues. First, the group of transformations leaving $[\cdot, \cdot]$ invariant comprises non-orthogonal elements and can be restricted, in the graphon case, only because of the canonical choice $F_Z = \text{uniform}[0,1]$ (a uniform probability measure not being available in $\mathbb{R}^d$). Distance-distorting transformations must be expected in general, as they are in the finite rank case (the matrices $\mathbf{Q}, \tilde{\mathbf{Q}} \in \mathbb{O}(p, q)$ in main text, Section 4). Second, the Wasserstein consistency criterion allows unboundedly high error in unboundedly high absolute numbers of nodes, provided their proportion vanishes, and this may break subsequent statistical analyses of the sort proposed here.

Figure 3: Graph regression. Different regression techniques are compared, with their achieved mean square error in brackets, on a 100-dimensional spectral embedding. The graph follows a latent position network model on $\mathbb{R}$ with kernel $f(x, y) = 1 - \exp(-2xy)$, and the embedding is therefore concentrated about a one-dimensional manifold, shown in Figure 1a). The responses follow a linear model $Y_i = a + bZ_i + \epsilon_i$. Further details in main text.

Figure 4: Kernel density ridge sets (red) as estimates of the underlying manifold (blue), for embeddings of simulated graphs described in Section 2 and also shown in Figure 1.

Figure 5: Non-linear dimension reduction of spectral embeddings. a) Graph of computer-to-computer network flow events on the Los Alamos National Laboratory network, from the publically available dataset [36], with colours indicating port preference; b) graph of computer-to-computer authentication events on the same network, with colours indicating authentication type; c) graph of consumer-restaurant ratings, showing only the restaurants, extracted from the publically available Yelp dataset, with colours indicating US state. In each case, the graph is embedded into $\mathbb{R}^{10}$, followed by applying Uniform manifold approximation [46].

Figure 6: Spectral embedding into $\mathbb{R}^2$. a) Graph of computer-to-computer network flow events on the Los Alamos National Laboratory network, from the publically available dataset [36], with colours indicating port preference; b) graph of computer-to-computer authentication events on the same network, with colours indicating authentication type; c) graph of consumer-restaurant ratings, showing only the restaurants, extracted from the publically available Yelp dataset, with colours indicating US state.