[Reviews · NeurIPS 2020]

Review 1

Summary and Contributions: This paper is motivated by the selection of the embedding dimension in latent space network models. In such models, there is typically a "true" model dimension $d$ (i.e., the dimension of the latent positions). When one applies a spectral embedding to a network generated from such a $d$-dimensional model, a key step is in choosing the dimension of this embedding, and it need not be the case that the optimal choice of this embedding dimension ($D$ in the present paper), because, for example, the model under study may not be low-rank, or the latent positions lie on a lower-dimensional manifold. This paper formalizes this notion, and shows that even when the true model dimension $d$ is large, the "true" spectral embedding dimension $D$ may be much smaller. This adds an interesting wrinkle to the question of model selection for network data-- many methods are concerned with correctly choosing $d$, but the results in this paper suggest that $D$ is the more appropriate target.

Strengths: This paper formalizes a notion that has become a sort of folklore among researchers working on latent space models. Given the centrality of graph data to contemporary machine learning, this paper will undoubtedly be of interest to a large portion of the NeurIPS community. The paper is exceptionally clear and well-written, with enough theoretical details included to give a sketch of the core ideas without overwhelming the reader with notation.

Weaknesses: My only complaints are to do with some minor presentation issues, which I have outlined in the "Clarity" field below.

Correctness: The results appear to be correct.

Clarity: The paper is exceptionally clear. The author(s) is/are to be commended. A few minor points follow. Assumption 3 seems to rule out the SBM (or any other model with a discrete component to its latent position distribution). If so, this should be discussed. Just before example 2 on page 5, it is not clear if the short paragraph spanning lines 137 and 138 is meant to pertain to Example 1 or to something else. Perhaps this sentence is misplaced? Also, if $d$ is the dimension of the latent space, isn't the implication of Theorem 3 that the points (resp. their estimates) lie on (resp. near) a set of dimension *other* than $d$, of have I misunderstood something? In the second display equation on page 6, the bound for the Laplacian embedding includes a sparsity term, but the bound for the ASE estimates does not. My impression was that a similar sparsity term should appear for both bounds. How was the 10-dimensional embedding chosen in the Los Alamos example? Of course this is ancillary to the point being made in the example, but in a paper of this sort, a connection to the model selection problem would be apt.

Relation to Prior Work: The work is well-situated in the context of other latent position models, that are amply cited. The author(s) might consider adding a citation to the more recent survey by Athreya et al on the RDPG (2018 in JMLR http://jmlr.csail.mit.edu/papers/volume18/17-448/17-448.pdf), which is a bit more comprehensive than the other citations for RDPG and its generalizations, and might consider a more substantial discussion of the latent space models studied by Hoff and coauthors over the years. The author(s) is/are clear about how these results relate to other recent works. Most similar to this seems to be Lei (2018, citation [33] in the manuscript), and the author(s) is/are appropriately modest about the fact that much of this paper is an extension of those results.

Reproducibility: Yes

Additional Feedback: Citation [13] (Caron and Fox's random measures paper) is used as an example of a model in which each node has a "sociability" score associated to it, that modulates degrees. It seems to me that it would be more apt to cite the degree-corrected SBM (or one of its variants) here. A few minor grammatical and typographical corrections follow: "Lower dimensional" should be hyphenated in the abstract. Machine learning need not be hyphenated in line 11 of page 1. Lines 13 and 14 should read "...from spectral embedding of a graph." The word "of" is currently missing.


Review 2

Summary and Contributions: This study proves that spectral embedding of latent position graphs lies in a low dimensional space.

Strengths: The claims seem sound and novel.

Weaknesses: I'm not sure the results attract much attention in the NeurIPS community.

Correctness: The claims seem correct.

Clarity: The paper is well witten.

Relation to Prior Work: Differences from previous contributions are well discussed.

Reproducibility: Yes

Additional Feedback: This study proves that spectral embedding of latent position graphs lies in a low dimensional space in the sense of Hausdorff dimension. The theoretical contributions are nice, but I'm worrying that the theory does not attract much attention in the NeurIPS community. In particular, all applications seem to have only loose connections to the main theory, plus, these applications do not provide any useful insights for practitioners. (1) Showing tighter relationships between the applications and the main theoretical results (e.g., why do the main results help make sense of t-SNE for visualization?) and (2) providing deeper insights about how the practitioners can decide the embedding dimension using the main results are helpful.


Review 3

Summary and Contributions: This paper studies spectral embedding of graphs. The main contribution is the demonstration that for certain graphs generated by graphons, the spectral embeddings will be (near) a low-dimensional manifold, even when the spectral dimension of the graph is infinite. ----- After reading the other reviews and the author response, I've changed my vote to accept.

Strengths: The high-level observation is new, interesting, and plausibly important. The techniques used to establish the result are new to me, and seem like they might be useful for future work. The main result itself also suggests some avenues for future work. The paper is fairly clearly written.

Weaknesses: The main weakness of the paper is that it's not totally clear how the theoretical results presented relate to or inform actual practice. However, that's true of basically all foundational work, and this paper is not particularly bad for this. However, I do think the motivation is a bit misleading and could be cleaned up. In particular, since graphon models are not realistic models for actual data, the observation that graphon-generated random graphs admit low-dimensional structure is not an explanation for why low-dimensional embeddings are successful in practice. Instead, I'd prefer if the contribution of the paper was cast as "for this theoretically tractable but reasonably rich model class, this behavior occurs, which may provide some insight as to why we witness similar behavior in real data". Related, the paper would benefit from a very brief discussion of the limitations of graphon models and how these relate to the interpretation of the results. For instance, does the fact that the generated graphs are almost surely dense matter? See, e.g., https://arxiv.org/abs/1312.7857 , https://arxiv.org/abs/1512.03099 , https://dl.acm.org/doi/abs/10.5555/3122009.3242067

Correctness: I believe the results are correct, though I did not check the math in detail

Clarity: The paper is clearly written

Relation to Prior Work: Yes, I felt that connections to previous work were well explained. However, I am not familiar with the literature on spectral embeddings.

Reproducibility: Yes

Additional Feedback: 1. at line 21, it would be good to clarify that all of the SBM variants are special cases of graphon models 2. Define the \perp operator subscript 3. The paragraph beginning at 67 feels misplaced. I wasn't sure what I was meant to get out of it, and it makes forward references to technical assumptions.


Review 4

Summary and Contributions: Motivated by the spectral embedding of an adjacency matrix, this paper studies spectral representations of a generalization of a graphon model, in which vertices are associated to a latent position on a d-dimensional space, and edge probabilities are given by a symmetric function f that takes as arguments latent position pairs, and thus the adjacency matrix is not necessarily low-rank (as it is commonly assumed). The authors present sufficient conditions under which this function f admits a representation in a manifold with Hausdorff dimension $d/\alpha$, where this $\alpha$ is somewhat related to the continuity of f. The authors provide examples in which they are able to calculate the dimension for the manifold, as well as particular models that can be represented with a function f with finite rank, and use the results of [50] to show consistency of the latent positions estimate under this scenario. Finally, the authors show some examples of subsequent inference problems for which a spectral embedding of the adjacency matrix can be successfully deployed.

Strengths: The paper presents interesting theoretical results. In Theorem 3, the authors provide an upper bound for the dimension of the manifold depending on the function f, which I think are novel and relevant for the community. The authors also provide examples of models under which f has finite rank, some of which are novel.

Weaknesses: Thanks to the authors for their response and for addressing my comments. My confusion about the infinite dimensional models has been clarified. I think it would help the readers if the authors define what is D in section 3. Also, as pointed out by Reviewer 1, the text in lines 137-138 is a bit confusing in that regard. ---------------------------------------------------- In my opinion, I think that the paper is limited in the study of the implications of the theoretical results. Sections 4 and 5 seem a bit disconnected from the new results introduced in the paper. In Section 4, the authors only focus on finite rank models, for which strong consistency results are already known [50], and leave the discussion of the difficulties in infinite rank to the supplementary material. I think it would be particularly interesting to know examples of classes of infinite rank models for which the authors are able to characterize the Hausdorff dimension from Theorem 3. I also wonder which classes of this infinite rank models satisfy assumption 2. The authors comment that this assumption is not the same as Holder continuity, but there is no discussion about how flexible this assumption is. In Section 5, the authors introduce some examples of subsequent inference with spectral embeddings, but there is no use of the theoretical results besides the high-level motivation that one can successfully embed into a sufficiently large dimension. I wonder if the authors can use the theory to give an upper bound on the dimension that is needed to embed a given model, and show how the subsequent inference performance is affected by this choice.

Correctness: The theoretical results seem to be correct. Although the empirical methodology is only used to illustrate the theory, the embedding dimension of the spectral embedding is arbitrarily chosen so I wonder if the authors can justify their choice in some way.

Clarity: The paper is clearly written and easy to follow in most parts, but there are a few notations that are confusing. For example, what is $\mathcal{P}$ in line 173?

Relation to Prior Work: The authors discuss the connection to existing work in finite dimensional models, and mention in line 36 that their work extends [33] but this point is not clearly discussed.

Reproducibility: Yes

Additional Feedback: The authors limit their discussion to models for which many results are already known (finite rank). In my opinion, I think it will be very interesting that the authors discuss implications of their new results at least in some examples of infinite rank models. Also, I suggest the authors to discuss if there are any practical implications of their theory when choosing an embedding dimension for a model.

[Author Response · NeurIPS 2020]

The author(s) would like to thank the reviewers for their time and expertise. This response will begin by discussing points shared across reviews, and will address the most critical reviewer-specific points as space permits.

*Infinite rank models.* It does not seem to have been made clear enough that Examples 1 (always) and 2 (typically) are models where the rank $D$ (embedding dimension) is *infinite* but the manifold dimension is finite (and equal to the latent position dimension $d$ in Example 1). This miscommunication is apparent, for example, in Reviewer 4's comment "it would be particularly interesting to know examples of classes of infinite rank models for which the authors are able to characterize the Hausdorff dimension from Theorem 3". Inspired particularly by the comments of Reviewers 1 and 4, the revision will contain a larger diversity of examples, extracted from Hoff's and co-authors' work, wherein spectral embeddings almost always fall in the "finite $d$, infinite $D$" regime. Reviewer 4 asks "which classes of [these] infinite rank models satisfy assumption 2" and for expanded discussion of the assumption's flexibility. The answer is somewhat hidden in Example 2. *A positive-definite kernel $f(x, y)$ (of arbitrary dimension, and of finite or infinite rank) satisfies Assumption 2 with $\alpha = 1$ if its Hessian exists and is bounded along the diagonal $x = y$.* To say the same for an indefinite kernel, one must replace $f$ with its (operator-sense) absolute value $f_+ + f_-$. This point will be added.

*Selection of embedding dimension.* The paper does *not* immediately offer an improvement to existing techniques. Nevertheless, in light of the reviewers' comments, there will be added discussion of this topic in the main text, which was previously delegated to the supplementary material and to reference [47]. The method by Zhu & Ghodsi (2006), which uses a profile-likelihood-based analysis of the scree plot, has for a long time provided a functional choice for many practitioners and is easily used within the R package igraph. For a theoretical treatment of dimension selection, the cases $D < \infty$ and $D = \infty$ must be distinguished. In the former, simply finding a consistent estimate of $D$ has limited practical utility: appropriately scaled eigenvalues of the adjacency matrix converge to their population value, and all kinds of unreasonable rank selection procedures are therefore consistent. But, to quote [47], "any quest for a universally optimal methodology for choosing the "best" dimension [...], in general, for finite $n$, is a losing proposition". In the $D = \infty$ case, reference [33] finds the appropriate rate under which to let $\hat{D} \to \infty$, to achieve consistency in the Wasserstein metric. Unlike the $D < \infty$ case, stronger consistency, i.e., in the maximum latent position error, is not yet available in the $D = \infty$ case — this is an ongoing and nontrivial effort. Reviewer 1 asks how the estimate $\hat{D} = 10$ was selected in Section 5.4. Here, because analysis is partly reproduced from [50], it seemed most expedient to abide by that paper's original choice, which was avowedly arbitrary. After computing the full spectrum overnight, the method of Zhu & Ghodsi (2006) actually returns an estimate $\hat{D} = 6$, so this will be reported.

*How theoretical results presented relate to or inform actual practice.* Reviewers 2 ("all applications seem to have only loose connections to the main theory"), 3 ("it's not totally clear how the theoretical results presented relate to or inform actual practice.") and 4 ("the paper is limited in the study of the implications of the theoretical results") would have liked to see more direct methodological applications of the theory. This will be addressed through the following revisions.

• Section 5.1 (Graph regression) — added discussion of achievable error rate. It may be conjectured that spectral embedding with $\hat{D} \to \infty$ appropriately slowly followed by neural network regression can achieve the rate $n^{-2\beta/(2\beta+d)}$, where $\beta$ is the regression function smoothness (in the Hölder norm) and $d$ is the *intrinsic* dimension of the data. This conjecture leverages recent work in reference [40] on the rate of neural networks under low intrinsic (Minkowski) dimension, and hopes of a strong consistency bound (mentioned above) under $D = \infty$.

• Section 5.3 (Visualisation A) — to be removed.

• Section 5.4 (Visualisation B) — to be renamed "Real data", include a new figure and report intrinsic dimension estimates. In anticipation of the comment "why do the main results help make sense of t-SNE for visualization?" (Reviewer 2), there was a brief remark in the original submission "Other methods such as Uniform manifold approximation were tested with comparable results". In retrospect, these results should simply have been presented instead, as they show similar information and are based on a concrete assumption that the data live close to a manifold of low dimension. These will now be included as an additional figure. Instead of selecting the embedding dimension, what the paper *does* allow is estimation of the latent position dimension $d$, on the basis of the intrinsic dimension of $\hat{X}_i$ (and assuming $\alpha = 1$ in Theorem 3), which can be estimated by several existing techniques (several implemented in Table 2, reference [50]).

Reviewer 2: "I'm not sure the results attract much attention in the NeurIPS community". Graphs have become some of the most studied objects in statistics/machine-learning and von Luxburg's tutorial on spectral clustering has over 8000 citations according to Google Scholar. In a nutshell, the present paper makes a case for "spectral manifold estimation" rather than "spectral clustering".

Reviewer 3: "I do think the motivation is a bit misleading [...]". The second sentence will be changed to "the object of this paper is show that, for a theoretically tractable but rich class of random graph models, such a phenomenon occurs in the spectral embedding of a graph."

Reviewer 3: "does the fact that the generated graphs are almost surely dense matter"? Sparsity *is* handled in the paper, if crudely, with a sparsity factor (Section 4 and Supplementary Material). However, from the suggested references it is clear that the reviewer has more sophisticated sparsity-inducing processes in mind, and these will be discussed.

[Meta-Review · NeurIPS 2020]

Three of the four reports are clearly positive. The only review that is somewhat critical explains its lower score by lack of direct applications. I agree, but do not see that as a problem: NeurIPS has a long history of publishing foundational work, the topic of the paper falls clearly within the remit of the conference, and foundational and theoretical work on embeddings is still rare; I would consider this work a welcome exception. The rebuttal promises a number of revisions; please make sure to add these in the final version.